# The Natural Fermentation of Greek Tsounati Olives: Microbiome Analysis

**DOI:** 10.3390/foods14152568

**Published:** 2025-07-22

**Authors:** Marina Georgalaki, Ilario Ferrocino, Davide Buzzanca, Rania Anastasiou, Georgia Zoumpopoulou, Despoina Giabasakou, Danai Ziova, Alexandra Kokkali, George Paraskevakos, Effie Tsakalidou

**Affiliations:** 1Laboratory of Dairy Research, Department of Food Science and Human Nutrition, Agricultural University of Athens, Iera Odos 75, 11855 Athens, Greece; ranastasiou@aua.gr (R.A.); gz@aua.gr (G.Z.); desgiampa@outlook.com.gr (D.G.); danaiziova@gmail.com (D.Z.); alexkokkalh@gmail.com (A.K.); et@aua.gr (E.T.); 2Department of Agricultural, Forest, and Food Science, University of Turin, Largo Paolo Braccini 2, Grugliasco, 10095 Torino, Italy; ilario.ferrocino@unito.it (I.F.); davide.buzzanca@unito.it (D.B.); 3International Probiotics Association, Los Angeles, CA 90035, USA; george@internationalprobiotics.org

**Keywords:** Tsounati, Mastoidis, Athinolia, Greek-style table olives, microbiological analysis, metataxonomic analysis, phenolics, triterpenic acids

## Abstract

The comprehensive analysis of microbial communities reveals the unique microbial identity of different olive varieties, paving the way for new strategies in their development and commercial exploitation. In this context, the present study aimed to explore the microbial diversity and functional characteristics of Tsounati variety olives from the Monemvasia region of Peloponnese, Greece, that were naturally fermented for three months. The bacterial and fungal microbiota of both olives and brines were fingerprinted throughout the fermentation through classical microbiological analysis combined with molecular techniques. Among the 148 isolated bacteria, 85 were lactic acid bacteria (LAB), and 63 belonged to the *Enterobacteriaceae* family, while the 178 fungal isolates comprised 136 yeasts and 42 non-yeast or yeast-like fungi. Metataxonomic analysis confirmed the dominance of the bacterial genera *Lactiplantibacillus*, *Leuconostoc*, along with the *Enterobacteriaceae* family, and it revealed the presence of *Coleofasciculaceae* cyanobacteria mostly in olives. The dominant fungal genera were yeasts, namely *Saccharomyces*, *Nakazawaea*, and *Cyberlindnera*. Using the Folin–Ciocalteu assay, the average total polyphenol content of Tsounati fermented olive samples was 761.80 ± 128.87 mg gallic acid equivalents kg^−1^ after 90 days of fermentation. The concentrations of the triterpenic, maslinic, and oleanolic acids, as determined by HPLC, remained stable throughout fermentation, with average values of 4764 and 1807 mg kg^−1^, respectively. Finally, sensory analysis revealed the rich aromatic character of Tsounati variety, highlighting its potential to be used for Greek-style table olive production.

## 1. Introduction

In the kingdom of Plantae, an outstanding family called *Oleaceae*, also known as the olive and lilac family, comprises 28 genera, one of which is extinct, and an estimated number of 700 species [1]. Notable members include ash trees, jasmine, and several popular ornamental plants, such as privet, forsythia, fringe trees, and lilac [2]. The genus *Olea* includes 12 species, with the most important one being, by far, the type species of the genus, namely olive (*Olea europaea* L.) [3]. This species is the only one that produces edible fruits, and it originates from the Middle East, where it was cultivated by 5000 BC, later spread to all countries bordering the Mediterranean Sea, and eventually reached America, Africa, Australia, New Zealand, and China [4,5]. To delve into the past, ancient DNA, trapped in the matrices of ceramic transport jars from Mediterranean shipwrecks, was sequenced, revealing the presence of olive fruits among other plants [6], while next-generation sequencing (NGS) technologies and bioinformatics have been applied to study the singular evolution of the *Olea* genome structure [7].

The number of olive varieties reported by *The World Catalogue of Olive Varieties* features 139 varieties from 23 olive-growing countries [8]. However, the cataloguing of the existing olive cultivars in the world has been a permanent challenge. To conserve the genetic heritage of the olive tree evolved over centuries through farmers’ selection, the International Olive Council (IOC) network of germplasm banks conserves cultivated varieties from around the world and currently holds more than 1700 accessions [9]. Nevertheless, different parameters determine their use (production of olive oil, table olives, or both), and only a few of them can be considered commercial [10], while highlighting the probable existence of unknown olive cultivars different from those characterized so far [11].

In Greece, according to the register of cultivated fruit tree varieties [12], a total of 64, namely 33 Greek and 31 foreign ones, are officially destined for table olive production. The total number of Greek cultivars exceeds 40 [9,13]. However, only nine Greek olive varieties, namely Adramitini, Amigdalolia, Halkidikis, Kalamon (or Kalamata), Konservolia (or Conservolea), Koroneiki, Mastoidis, Megaritiki, and Valanolia, are included in *The World Catalogue of Olive Varieties*, which has been compiled under the guidance of IOC [8,14].

Although the most significant Greek olive varieties are Kalamon, Konservolia, and Halkidikis, there are others with outstanding organoleptic characteristics. The Mastoidis variety (*Olea europaea* var. *mamilaris*), which is officially reported as Tsounati in The National Catalogue of Olive Varieties but is also called “Athinaiki” or “Athinolia”, is one of the most important varieties for the economy of the Peloponnese region. It is cultivated in South Laconia as well as in Argolida and Western Crete [15,16,17]. The fruit weight is 137–141 g per 100 olive fruits, while its oil concentration of 507.8–517.3 g kg^−1^ dry mass makes it ideal for olive oil production [18].

According to the IOC [19] for the crop year 2022–2023, Greece ranked second in table olives exports with 188,861 tons, valued at approximately EUR 739.5 million, trailing behind Spain with 267,142 tons valued at EUR 1005.2 million. These data highlight the significance of table olives in the Greek economy, establishing Greece as a key player in global olive exports. Moreover, table olives are an undoubtable nutritional treasure of high energy content (180–250 kcal/100 g), unsaturated fatty acids, proteins containing essential amino acids, and functional phytochemicals, such as phenolic compounds and triterpenic acids [20,21]. Multiple health benefits have been associated with their consumption with respect to cardiovascular health, protection from oxidative damage, and anti-inflammatory activity [21]. Moreover, multiple biological effects of triterpenic acids both in vitro and in vivo have been reported [22,23].

In Greece, the most used practice for table olive production is the Greek-style natural fermentation of mainly black olives immersed in brine without any debittering treatment [10]. Due to brine’s high salt concentration (6–10%), the spontaneous fermentation process is mainly driven by yeasts and also by lactic acid bacteria (LAB), as well as Gram-negative bacteria that dominate mostly the first phase of the fermentation.

The complex and amazing biosystem of Greek-style table olives has been studied not only by culture-dependent approaches but by metataxonomics as well. There are numerous references concerning the microbiome of natural green olives, belonging to diverse varieties, such as Nocellare etnea [24,25], Bella di Cerignola [26], Aloreña de Málaga [27,28], Halkidikis [29], Picual [30], and Manzanilla [31]. Fewer studies on fermented natural black olives, belonging to Konservolia [29], Nyons [32], and Kalamata [30,33] varieties, have been also reported. Additionally, the bacterial diversity of commercial table olive biofilms belonging to various green and black olive varieties has been explored by metataxonomic and compositional data analysis [34].

Tsounati is a Greek olive variety that, so far, is exclusively used for the production of high-quality olive oil. The objective of the present study was to address the challenge of determining, for the first time, whether Tsounati-variety olives can also be utilized for the production of high-quality table olives. The fermentation process was monitored through both physicochemical and microbiological analyses, while the final product was evaluated from a sensory perspective.

## 2. Materials and Methods

### 2.1. Olives Fermentation

Olives of the Tsounati variety were collected from the Monemvasia region of Peloponnese, Greece, in November of the 2022–2023 harvest period (at the green stage of maturation). Drupes were transferred to the laboratory and directly washed with tap water and carved with a knife (split) by scoring in three places lengthwise without reaching the core, to facilitate the removal of their bitter taste. Three technical replicates of Greek-style natural fermentation (Fermentations A, B, and C) were performed. Olives (3 kg per fermentation) were placed in three different plastic containers of 5.0 L volume, and these were filled with brine (2 L, 6% *w*/*v* NaCl). Fermentation took place for 90 days at room temperature (~25 °C). Olive and brine samples were collected at various time points (0, 4, 7, 15, 30, 60, and 90 days) and stored at −80 °C until further analyses. Classical microbiological analysis was performed on the day of sampling. The sample coding of olives and brines is presented in Appendix A.

### 2.2. Physicochemical Analysis of Olives and Brines

#### 2.2.1. pH and Salt Content

The pH of brine samples was measured using a pH meter (827 pH lab, Metrohm, Herisau, Switzerland) throughout the fermentation period. Salt concentration (NaCl, % *w*/*v*) of the brine samples was measured every seven days using a Beaumé hydrometer (Alla, Chemillé en Anjou, France) and expressed as Beaumé degrees. During the fermentation process, brine salt concentration was adjusted by the periodic addition of salt to 6% (*w*/*v*), which represents the minimum NaCl content of naturally fermented olives [35].

#### 2.2.2. HPLC Analysis of Sugars, Organic Acids, Alcohols, and Triterpenic Acids

In brine samples, sugars, organic acids, and ethanol were determined by HPLC analysis (GBC Scientific Equipment Pty Ltd., Dandenong, Victoria, Australia) on an HPX-87H Aminex column (BioRad, Hercules, CA, USA) according to Angelopoulou et al. [36]. Triterpenic acids were determined in olive and brine samples by HPLC analysis using the same equipment on a Spherisorb ODS-2 column (Waters Inc., Mildford, MA, USA) as described by Alexandraki et al. [20]. The extraction of triterpenic acids from the flesh of olive samples as well as from brine samples was performed as described by Romero et al. [37], with slight modifications in the case of olive samples [20]. Pure maslinic and oleanolic acids (Sigma, St. Louis, MO, USA) in a concentration range of 0–3000 mg L^−1^ were used as standards for standard curve plots and quantification. Each sample was measured in duplicate, and the results were expressed as mM mL^−1^ brine.

#### 2.2.3. Total Phenolic Content

The Folin–Ciocalteu assay [38] was performed to measure TPC in both olive and brine samples.

Regarding brine samples, the method of Marsilio et al. [39] was followed using 0.1 mL of brine sample. More specifically, 1.0 mL of Folin–Ciocalteu reagent (Carlo Erba Reagents, Val de Reuil Cedex, France) was added to each brine sample. After vortexing, samples remained still for 5 min, and 3.0 mL of 20% *w*/*v* Na_2_CO_3_ solution (Carlo Erba Reagents) was added. The volume of the samples was adjusted to 10 mL with deionized water, and samples were left undisturbed for 30 min.

The same procedure was performed for the olive samples, which were previously prepared as described by Romero et al. [37], with slight modifications [20]. Samples were finally centrifuged at 3500× *g* for 2 min, and supernatants were transferred to new tubes. Blanks were prepared by mixing 1.0 mL of Folin–Ciocalteu reagent, 3.0 mL of 20% *w*/*v* Na_2_CO_3_ solution, and 6.0 mL of deionized water.

Brine and olive samples were measured photometrically (Lambda 20 UV/Vis Spectrophotometer, PerkinElmer, Norwalk, CT, USA) at 725 nm. TPC was determined using a calibration curve created with gallic acid solutions (Sigma-Aldrich, St. Louis, USA) of known concentrations ranging from 0 to 3 mg mL^−1^. Each sample was measured in duplicate, and the results were expressed as gallic acid equivalents (GAE mg kg^−1^ olives or mg mL^−1^ brine).

### 2.3. Microbiological Analysis of Olives and Brines

Olive samples (10 g of olive flesh) were homogenized for 1 min using 90 mL of sterilized Ringer’s solution (Biokar Diagnostics, Beauvais, France) in a stomacher (Lab-Blender 400, Seward Ltd., London, UK). Homogenized olive and brine samples were serially diluted in the aforementioned medium. The following groups of microorganisms were enumerated throughout the fermentation at days 0, 4, 7, 15, 30, 60, and 90: (1) total mesophilic bacteria on Plate Count agar (PCA, Biokar Diagnostics) at 30 °C for 3 days; (2) mesophilic lactobacilli on De Man–Rogosa–Sharpe agar (MRS, Biokar Diagnostics) with a pH of 6.4 containing cycloheximide (50 μg mL^−1^) at 30 °C for 3 days; (3) *Enterobacteriaceae* on Violet Red Bile Glucose agar (VRBG, Biokar Diagnostics) at 37 °C for 24 h, under microaerophilic conditions (double agar layer); and (4) fungi on Yeast Glucose Chloramphenicol agar (YGC, Condalab, Madrid, Spain) at 25 °C for 5 d. Μicrobial counts of olives (log CFU g^−1^) and brines (log CFU mL^−1^) were expressed as mean ± standard deviation (SD) values (*n* = 2).

Different colonies were collected from MRS and YGC agar plates (used for enumerating mesophilic lactobacilli and fungi, respectively) based on their morphology, i.e., shape, size, and color. They were purified by repetitive streaking and finally stored for further study at −80 °C in MRS broth (bacteria) or liquid medium with 0.5% *w*/*v* yeast extract and 2.0% *w*/*v* glucose (fungi), containing 20% *v*/*v* glycerol.

### 2.4. DNA Extraction and Rep-PCR Fingerprinting of Bacterial and Fungal Isolates

Total bacterial DNA was extracted from 2 mL of fresh pure MRS cultures of all isolates at the exponential phase, following a protocol previously reported [40]. DNA extraction from fungal pure cultures grown overnight in medium containing 5% *w*/*v* yeast extract and 20% *w*/*v* glucose was performed according to the protocol of Kopsahelis et al. [41]. However, cell pellet was washed twice with dd H_2_O and incubated at 65 °C for 10 min before lysis, aiming at the partial inactivation of deoxyribonucleases (DNases). Ice-cold 70% *v*/*v* ethanol was used to wash the DNA pellet, which was finally resuspended in 30–50 μL of Tris-EDTA (TE) buffer (10 mM Tris-HCl, 1 mM EDTA, pH 8.0). DNA concentration was measured using a NanoDrop One Microvolume UV spectrophotometer (ThermoFisher Scientific, Madison, WI, USA).

Repetitive extragenic palindromic elements-PCR (rep-PCR) was performed for fingerprinting of the isolates using a SimpliAmp™ Thermal Cycler (ThermoFisher Scientific, Waltham, MA, USA). The protocol of Georgalaki et al. [40] was employed for bacteria. The protocol of Silva-Filho et al. [42], slightly modified, was employed for fungi. More specifically, PCR reaction volume (25 µL) contained 50 ng of DNA, 0.3 mM (GTG) of 5 primer (5′-GTG GTG GTG GTG GTG-3′; VBC Biotech, Vienna, Austria) and 12.5 µL of OneTaq-Quick Load 2 × Master Mix (New England Biolabs, Ipswich, MA, USA). Amplification was performed as follows: initial denaturation at 94 °C for 5 min, 35 cycles with denaturation at 94 °C for 15 s, primer annealing at 55 °C for 45 s and primer extension at 72 °C for 90 s, followed by a final extension at 72 °C for 15 min. Separation of bacterial and fungal rep-PCR products was electrophoretically performed, and the BioNumerics version 6.0 software (Applied Maths, Ghent, Belgium) was used for rep-PCR fingerprint clustering [40].

### 2.5. Identification of Bacterial and Fungal Isolates by 16S and ITS rDNA Sequencing

The selection of representative bacterial and fungal isolates was based on the clustering of the rep-PCR analysis. Identification at the species level of MRS and YGC isolates was performed by 16S rDNA sequencing [43] and ITS rDNA sequencing [44], respectively. The primers used were 16SF (5′-GGA GAG TTA GAT CTT GGC TCA G-3′)/16SR (5′-AGA AAG GAG GTG ATC CAG CC-3′) and ITS1 (5′-TCC GTA GGT GAA CCT TGC GG-3′)/ITS4 (5′-TCC TCC GCT TAT TGA TAT GC-3′), respectively. NucleoSpin^®^ Gel and PCR Clean-up (Macherey-Nagel, Duren, Germany) were further used for DNA purification after electrophoresis. Identification at the species level was achieved using the Basic Local Alignment Tool available at the NCBI (National Center for Biotechnology Information (https://blast.ncbi.nlm.nih.gov/Blast.cgi? (accessed on 8 April 2024)).

### 2.6. Total DNA Extraction and Metataxonomic Analysis

Olive pulp samples were prepared according to Kazou et al. [33]. The DNeasy^®^ PowerSoil^®^ Pro Kit (Qiagen, Valencia, CA, USA) was used for total genomic DNA extraction and purification from olive pulp samples (1 g of olive pulp pellet obtained from 12 g of olive pulp). The DNeasy^®^ PowerFood^®^ Microbial Kit (Qiagen) was used for total genomic DNA extraction and purification from brine samples (15 mL) according to the manufacturer’s instructions. A NanoDrop^TM^ One Microvolume UV Spectrophotometer (ThermoFisher Scientific) was used for DNA content measurement before sequencing. Evaluation of the bacterial and fungal microbiota was performed by amplicon sequencing (bTEFAP^®^) performed at Molecular Research DNA (MR DNA, Shallowater, TS, USA) on the Illumina MiSeq. Evaluation of the bacterial diversity was performed using the primers 27F (5′-AGR GTT TGA TCM TGG CTC AG-3′)/519R (5′-GTN TTA CNG CGG CKG CTG-3′), targeting the V1–V3 hypervariable region of the 16S rRNA gene. Evaluation of the fungal diversity was performed using the primers ITS1F (5′-CTT GGT CAT TTA GAG GAA GTA A-3′)/ITS2R (5′-GCT GCG TTC TTC ATC GAT GC-3′), targeting the internal transcribed spacer (ITS) DNA region, namely ITS1-ITS2 [33]. PCR and purification of amplicon products were performed according to Papademas et al. [45].

### 2.7. Bioinformatic and Statistical Analysis

Raw reads were imported in QIIME2 software (vr. 2022.2.0, https://docs.qiime2.org/ (accessed on 10 February 2025)) [46] for denoising by the script qiime dada2 denoise-paired [47]. The amplicon-sequence variants (ASVs) obtained were then used for taxonomic assignment against the greengenes2 [48] or UNITE database for bacteria and fungi, respectively, and rarefied at the lowest number of sequence/sample (43,589 for bacteria and 10,194 for fungi). Alpha and beta diversity calculations were performed by the diversity script of QIIME2. Statistical analysis about taxa was performed with Microeco vr. 1.4.0 [1] in RStudio vr. 2023.09.1 (R vr. 4.4.2; 2024-10-31 ucrt). Homogeneity tests were performed in R (vr. 4.4.2; 2024-10-31 ucrt) using Shapiro–Wilk’s W and Modified Levene’s tests (Brown–Forsythe test). The Kruskal–Wallis test followed by Dunn’s post hoc test and ANOVA followed by Tukey’s post hoc test were used for non-parametric and parametric data, respectively.

The amplicon sequencing raw sequence reads are available at the NCBI (https://www.ncbi.nlm.nih.gov/bioproject/ (accessed on 10 February 2025)) bioproject number PRJNA1222373.

### 2.8. Sensory Evaluation of Olives

The method of sensory analysis of table olives established by the IOC [49] was used for the sensory evaluation of olive samples. A taste panel consisting of 10 trained people evaluated the samples, while olives were well mixed in the containers prior to their presentation in the tasting glass plates to ensure sample representativeness. An evaluation sheet was used to fulfill the scores, and the intensity ranged from 1 (low level) to 11 (high level). The panel members determined the intensities of the attributes listed in the profile sheet, namely smell, color, abnormal fermentation, salty, bitter, acid, hardness, fibrousness, and crunchiness. Two commercial, naturally fermented green olive samples of Konservolia variety were included in the sensory evaluation.

### 2.9. Statistical Analysis of Experimental Values

Data were expressed as mean ± SD values per sample (*n* = 6; three fermentations A, B, and C, and two technical repetitions for each fermentation). Two-sample comparisons were performed with Student’s *t*-test. Multiple-sample comparisons were performed with one-way analysis of variance (ANOVA) and post hoc Tukey’s. Differences were considered statistically significant for *p* < 0.05. All statistical procedures were carried out using Statgraphics^®^ Centurion^TM^ XVII software (StatPoint Technologies Inc., Warrenton, VA, USA).

## 3. Results and Discussion

### 3.1. Physicochemical Analysis of Olives and Brines

#### 3.1.1. pH and Salt Content of Brines

Regarding brine samples, the initial pH value was 7.30 ± 0.00. It decreased to 4.91 ± 0.11 and 4.65 ± 0.60 at days 4 and 7, respectively, and remained thereafter stable at the value of approximately 4.73 ± 0.30 until the end of the fermentation period. The pH value of 4.73 is higher than the pH values previously reported (≤4.50) for Greek-style olives at the end of fermentation [29,32]. Moreover, it does not comply with the physicochemical characteristics for the safety of the final product that report a maximum pH value of 4.30 [35]. Actually, multiple parameters determine the dynamics of fermentation, namely indigenous olive microbiota itself, variety, degree of ripeness, region, and farming practices [50]. Additionally, the high presence of phenolic compounds in the brine incommodes LAB growth, which is related to the pH decrease in directly brined olives [51]. It is reported that Manzanilla fermented olives, affected by butyric, sulfidic, or putrid spoilage, had pH values > 4.60, 7.70, and 4.80, respectively, while samples following unspoiled fermentation had pH values ≤ 4.3 [52]. The potential risks associated with spoilage and food safety are considerable due to the high pH value, which may permit the growth of bacteria, such as *Enterobacteriaceae*. However, the 6% *w*/*v* NaCl concentration in the brine inhibits the development of abnormal fermentations, as it results in a water activity (aw) of 0.964, a value approaching the tolerance limit for *Enterobacteriaceae* (0.93) [53]. Additionally, the high concentration of phenolic compounds in the brine further suppresses the growth of *Enterobacteriaceae*, due to their antimicrobial properties [54].

Sodium intake has been associated with various health outcomes, including blood pressure regulation, cardiovascular disease risk, and bone health. Consequently, numerous studies have investigated the effects of sodium consumption [55]. In addition, the European Food Safety Authority (EFSA) Panel on Nutrition, Novel Foods, and Food Allergens (NDA) established dietary reference values (DRVs) for sodium, as requested by the European Commission, recommending a total daily salt intake of less than 5.0 g, equivalent to a sodium intake of less than 2.0 g per day. As reported by IOC, in 2019, in the EU, the annual per capita consumption of table olives ranged between 0.3 and 4.4 kg (corresponding to 0.8 and 12.1 g olives per capita per day), depending on the country (https://www.internationaloliveoil.org/wp-content/uploads/2021/02/NEWSLETTER_IOC-160_EN.pdf (accessed on 4 June 2025)). Cyprus was the leading consuming country (4.4 kg per capita annually), followed by Spain, the world’s top producer (4.1 kg), Luxembourg (2.7 kg), Bulgaria (1.8 kg), and Malta (1.7 kg). Further down the ranking, Greece had an annual consumption of 1.5 kg per capita. In the present study, the salt concentration in the olives was not directly determined; however, considering that the NaCl concentration in the brine throughout the fermentation was adjusted to 6% *w*/*v*, it can be hypothesized that the salt content in the olives did not exceed this value and therefore remains within the recommended limits for NaCl intake set by EFSA [55] and the World Health Organization (WHO 2023, https://www.who.int/news-room/fact-sheets/detail/salt-reduction (accessed on 4 June 2025)).

#### 3.1.2. HPLC Analysis of Sugars, Organic Acids, Alcohols, and Triterpenic Acids

In the present study, sugars, organic acids, and ethanol were determined by HPLC in brine samples. These compounds are either naturally present in the olive fruits or are formed during fermentation in brine. Additionally, sugar diffusion takes place from olives to brine, and an equilibrium takes place after approximately seven days of fermentation, as previously reported [56]. HPLC analysis revealed the presence of glucose, fructose, acetic acid, tartaric acid, glycerol, 2,3-butanediol, and ethanol (Appendix A).

Glucose was detected on day 4 of fermentation, as expected because of its diffusion from the olives to the brine. It was then catabolized by bacteria and fungi and was not further detected. Its average concentration of 53.42 ± 10.23 mM was similar to that reported (45 mM) by Anagnostopoulos et al. [57] for green naturally fermented Cypriot-variety olives. Fructose was detected for the first time on day 7 of fermentation. It increased until day 15 (*p* < 0.05) and then decreased or remained stable, depending on the fermentation, until the end of fermentation. The maximal average fructose concentration of 54.33 ± 3.80 mM on day 15 was higher than the concentration reported by Anagnostopoulos et al. [57], who determined 12 mM after 20 days of fermentation of green Cypriot naturally fermented olives. The total depletion of fructose, reported by Anagnostopoulos et al. [57], was not observed, probably due to the low presence of fructophilic fungal species. The fermentation of both sugars is attributed to microbial catabolism via the glycolysis pathway [58], more specifically the activity of homofermentative, as well as heterofermentative or facultatively heterofermentative, LAB, e.g., *Lactococcus* spp., *Leuconostoc* spp., and *Lactiplantibacillus* spp., respectively. Additionally, fungi that grow in fermented olives are also involved in sugar fermentation [59].

Acetic acid was detected on day 4 and increased until the end of fermentation (*p* < 0.05), as expected due to the action of heterofermentative LAB and fungi, mostly yeasts, which can oxidize acetaldehyde and ethanol produced during the alcoholic fermentation [60]. Its maximal average concentration of 16.00 mM ± 1.23 on day 90 was similar to the concentration reported by Ruiz-Barba et al. [31], who measured 20 mM after 90 days of fermentation of green Manzanilla Spanish-type olives, but lower than 220 mM, the concentration reported by Anagnostopoulos et al. [57] in naturally fermented green olives after 100 days of fermentation. However, similar concentrations were measured in fermentations performed with starter cultures [57]. Tartaric acid, which is a chemical compound of fresh olives, was detected in the brines from day 7 to the end of fermentation (*p* > 0.05). Its average concentration (22.21 ± 2.69 mM on day 90) was higher than previously reported, e.g., 8.4 mM [56] and 8 mM [57], after 70 days of fermentation. Moreover, only traces of lactic acid were detected, probably due to its conversion to acetic acid by LAB [61] and/or to its assimilation by some fungal species [62].

Ethanol was first detected on day 4, increased until day 60 (*p* < 0.05), and was further stable (maximal average concentration 56.21 ± 2.68 mM on day 60). Its increase can be attributed to the fungal, mostly yeast, metabolism, the bacterial heterofermentative fermentation, as well as to the catabolism of citric acid by LAB and fungi [63]. Similar or higher concentrations have been previously reported, e.g., 40 mM after 90 days of fermentation of green Manzanilla Spanish-type olives [31] and 250 mM after 90 d of fermentation of Cypriot olives [57]. Glycerol was first detected on day 4 and increased until day 90 (*p* < 0.05). Its presence is credited to fungal metabolism, as it acts as an osmoregulatory factor and a reserve of NADH, which is the necessary co-enzyme for “anaerobic” glycolysis [64]. The maximal average concentration of 57.82 ± 4.74 mM detected on day 90 was lower than previously reported, e.g., 200 mM in green naturally fermented Cypriot olives after 100 days of fermentation [57]. Lower concentrations could be attributed to the olive variety, as well as the low salt concentration of the brine, which influences the growth of LAB [65,66,67]. Finally, 2,3 butanodiol was detected from day 4 to the end of fermentation (*p* < 0.05); however, its average concentration on day 90 (1.74 ± 0.18 mM) was lower than previously detected, e.g., 15 mM on day 12 of the fermentation of green Manzanilla olives of the Spanish type, probably due to the different variety and the lower salt concentration of the brine [68]. The presence of 2,3-butanediol is attributed to the catabolism of citric acid by LAB, as well as to the metabolism of members of the *Enterobacteriaceae* family [68].

The concentration of triterpenic acids was measured in olives before and throughout fermentation (Appendix A). In untreated olives, the average concentrations of maslinic and oleanolic acids were 5041.73 ± 398.13 and 1672.18 ± 174.60 mg kg^−1^, respectively. The concentrations of both acids were consistently preserved throughout fermentation, despite high standard deviations observed for both acids’ concentrations, probably due to variations during the extraction procedure. The average value of maslinic acid was 4764.53 mg kg^−1^ and values ranged between 4854.35 (day 0) and 4663.46 mg kg^−1^ (day 90), while the average value of oleanolic acid was 1807.60 mg kg^−1^ and values ranged between 1689.84 (day 0) and 1966.13 mg kg^−1^ (day 90). Dissimilar values determined among different time points were not statistically significant (*p* > 0.05), supporting the fact that the concentrations of both triterpenic acids were not affected by the fermentation process [20].

The above-mentioned values are similar to or higher than values reported in the literature for table olives at the end of fermentation. Values reported for maslinic acid range from 1260 (Kalamon variety) to 2500 mg kg^−1^ (Arbequina variety), and values reported for oleanolic acid range from 366 (Cellina di Nardò variety) to 900 mg kg^−1^ (Empeltre variety) [20,69,70]. These variations may be attributed to multiple factors, which may influence the composition of table olives, e.g., olive variety, geographical location, agronomical and technological practices, fruit ripeness, and crop season [71].

It has already been reported that the natural processing of Greek table olive production (Kalamata and Conservolea) did not decrease the triterpenic acid content in the final product [20]. Additionally, in the present study, maslinic acid content was higher than that of oleanolic acid, which is in agreement with the data previously reported [20,37,69,72]. Finally, as formerly stated [69], no triterpenic acids were detected in the brine samples, as they are not dissolved in brine when the pH value is lower than 7.0 [37].

Triterpenic acids offer multiple health benefits, exhibiting cardioprotective, neuroprotective, anti-tumor, anti-diabetic, anti-oxidant, anti-hypertensive, anti-hyperlipidemic, anti-viral, and anti-inflammatory actions [22,23,73,74]. The detection of high concentrations of both maslinic and oleanolic acids throughout the fermentation process highlights the potential of the Tsounati variety for table olive production and positions it as a natural dietary source of these compounds.

#### 3.1.3. Total Phenol Content of Fermented Olive and Brine Samples

Concerning olive samples, TPC at the beginning of fermentation was rather low (average value 1079.11 ± 58.08 mg GAE kg^−1^) compared to that of other olive varieties previously reported (Appendix A). Anagnostopoulos et al. [57] reported approximately 5500 mg GAE kg^−1^ at the beginning of fermentation for the green Cypriot cracked olives. It is established that the chemical composition of olives, including TPC, is determined by genetic factors of the variety, as well as by other factors, i.e., climate, size and maturity stage of the fruit, harvest time, and adopted cultivation procedures [75,76,77]. It is worth mentioning the importance of irrigation, not only in the reduction of the pigments, alpha-tocopherols, and oleic acid, but also in the total phenol compound content [78]. García et al. [79] reported that the content in phenolic compounds of black processed olives was higher in fruits from non-irrigated than irrigated trees, particularly those of hydroxytyrosol, tyrosol and luteolin 7-glucoside. Olives used in the present study were derived from daily irrigated olive trees. Thus, we assume that irrigation rather than variety may be the reason for the low TPC of untreated olive samples in the beginning of fermentation.

The average TPC value of Tsounati fermented olive samples was 761.80 ± 128.87 mg GAE kg^−1^ after 90 days of fermentation (Appendix A). Higher TPC values ranging from 1000 to 2410 mg GAE kg^−1^ have been previously reported for both green and black naturally fermented olive samples at the end of fermentation (green cracked Cypriot olives) [57] (industrially produced black Kalamata olives [80]; Kalamata black olives [81]).

It is established that TPC tends to decrease during natural fermentation, not only because of the occurring diffusion to the brine, but also because of the hydrolysis of olives’ indigenous phenolic compounds, such as oleuropein, driven by endogenous or microbial enzymes. Concerning bacteria, the activity of *L. plantarum* strains, which can cause the degradation of phenolic acids to their corresponding vinyl derivatives by phenolic acid decarboxylase activity has been reported [76,82]. Additionally, yeasts dominating the microbiome of fermented olives may exhibit high enzymatic activity in the degradation of phenolic compounds and therefore play an important role in the olive debittering process [57,83]. However, the hydrolytic degradation of oleuropein is significantly slower in naturally fermented olives than in olives treated with a sodium hydroxide solution [84]. The detection of a significant loss of TPC during fermentation (*p* < 0.05) has been previously reported for naturally fermented green olives, e.g., Carolea variety [85], as well as black olives, e.g., Kalamata variety [81]. Moreover, as TPC is influenced by the condition of the olives in the beginning of fermentation, a low content of TPC would be expected at the end of fermentation because of the drupe-splitting procedure, which was performed. The cutting into the olive skin lengthwise of the fruit facilitates brine entering but also leads to a higher lixiviation of phenolic compounds to the brine [77]. However, in the present study, a low decrease (29.4%) of TPC was observed during the fermentation period. The fact that TPC did not suffer a high loss during fermentation could be attributed to lower microbial activity affecting phenolic compounds, although further experiments should be performed to verify this hypothesis. Nevertheless, although the final phenolic content of olives is affected by a plethora of factors, it is sufficiently large enough to protect the oil in the fruit against oxidation during storage [75].

The TPC of all brine samples increased gradually during the fermentation, as also reported by Anagnostopoulos et al. [57]. The TPC average value was 0.85 ± 0.12 mg GAE mL^−1^ brine at day 4 of fermentation and reached the highest average value 2.63 ± 0.08 mg GAE mL^−1^ brine on day 60 (Appendix A). Slightly lower values were measured on day 90. The increase of TPC in brines was expected as phenolic compounds, including oleuropein, are diffused from olives to brines during fermentation [25,79]. The decrease of TPC observed in brines after 60 days of fermentation may be attributed to the hydrolysis of phenolic compounds, such as oleuropein, driven by endogenous olive enzymes and microbial enzymes, such as esterase and β-glucosidase, which can convert complex phenolic compounds into simpler forms, e.g., hydroxytyrosol and elenolic acid [75,84]. The values observed for the Tsounati brine samples at the end of the fermentation are similar to values previously reported for the brine samples of green fermented olives, ranging from 0.55 to 3.7 mg mL^−1^ (Moroccan olives [76]; Nocellara Etnea variety [25]; cracked Cypriot olives [57]). Similar or even higher values (2.27 to 6.29 mg GAE mL^−1^ brine) have been reported for naturally fermented black olives (Kalamata olives [81]).

It is worth noting that, in large-scale fermentations, differences in pH reduction and phenolic content may be observed compared to laboratory-scale ones. These variations are attributed to greater fluctuations in salt concentration, temperature, and microbial distribution within larger tanks [53]. More specifically, polyphenol oxidation can be accelerated at low salt concentrations, since the solubility of oxygen species in saline solutions increases with decreasing salinity and rising temperature. Therefore, careful selection and use of microbial starters, along with the optimization of processing parameters, such as fermentation temperature, are crucial for achieving the desired fermentation outcomes on an industrial scale.

### 3.2. Microbiological Analysis of Olives and Brines

Results of the classical microbiological analysis of olive and brine samples are summarized in Appendix A.

The microbiological analysis of untreated Tsounati olives, before immersion in tap water, resulted in the following microbial populations: total mesophilic bacteria 7.76 ± 0.06, presumptive mesophilic LAB 7.09 ± 0.34, fungi 6.51 ± 0.25, and enterobacteria 7.41 ± 0.14 log CFU g^−1^. These counts indicate the high initial microbial load of olives before fermentation. In fermented olive samples (Appendix A), all microbial groups exhibited stable growth up to day 15 of fermentation (*p* > 0.05). Until day 30, total mesophilic bacterial counts and presumptive mesophilic LAB decreased less than 2 logs, fungal counts showed a decrease of less than 1 log, while, for enterobacteria, a 4 log reduction was determined (*p* < 0.05). Afterwards, no significant differences among fermentation time points were observed for all microbial groups (*p* > 0.05), which plateaued until day 90, except for enterobacteria that were eliminated between day 30 and 60 of fermentation.

Regarding brine samples (Appendix A), all microbial populations, except that of enterobacteria, increased gradually from the beginning (day 0) until day 7 of fermentation (*p* < 0.05) reaching maxima counts of 7.79 ± 0.17 (total mesophilic bacteria), 7.75 ± 0.17 (presumptive mesophilic LAB), and 6.61 ± 0.25 (fungi) log CFU mL^−1^. Their growth was then gradually decreased, reaching 5.78 ± 0.13, 5.68 ± 0.40, and 6.09 ± 0.22 CFU mL^−1^, respectively, at day 30 (*p* < 0.05) and further increased, reaching counts of 6.27 ± 0.27, 6.16 ± 0.19, and 6.22 ± 0.16 log CFU mL^−1^, respectively, at the end of fermentation. The growth of enterobacteria increased during the first four days of fermentation (*p* < 0.05) and then decreased by less than 1 log until day 15 (*p* < 0.05). However, it dropped drastically after 30 days of fermentation (1.35 ± 0.34 log CFU mL^−1^) and was eliminated by day 60.

The results obtained for microbial counts suggest no significant differences among the growth patterns of the bacterial and fungal populations throughout the fermentation of Tsounati olives. They also indicate their parallel growth until day 90 and an equilibrium between olives and brines, taking place from day 7 until the end of the fermentation (day 90). The growth of fungi was expected, as they can not only grow on residual sugars, reaching population levels above 6 log CFU mL^−1^ and causing brine clouding [86], but they can also tolerate acidic environments with pH values around 3.5 [87]. Additionally, they can utilize lactic and acetic acids that are produced during fermentation as carbon and energy sources [88], while tolerance to phenolic compounds depends on their chemical nature, as well as the fungal species and strain [89]. The growth of mesophilic LAB and fungi influenced the population of enterobacteria and resulted in their final disappearance on day 60, as it was expected due to pH decrease. Indeed, there are multiple references concerning Gram-negative bacteria, such as *Enterobacteriaceae*, found at the beginning of the olive fermentation and decreased after a few weeks of fermentation due to the acidification of the environment by LAB [90,91,92]. On the other hand, there are references reporting that the *Enterobacteriaceae* population was even lower than the detection limit of the method (<1 log CFU g^−1^) from the beginning of olive fermentation [29].

### 3.3. Rep-PCR Fingerprinting and Identification of Bacterial Isolates

A total of 148 bacterial isolates (MRS agar plates, 30 °C) were collected from olives (109) and brines (39) (Table 1 and Appendix A).

Based on the rep-PCR analysis, bacterial isolates were clustered in 48 groups (Appendix A). The 16S rDNA sequencing was performed for selected representative isolates of all rep-PCR groups. However, no rep-PCR products were obtained for 11 isolates, namely, 93a, 141, 142, 155, 160, 171, 195, 226, 233, 241, and 249; therefore, since they could not be clustered, they were also identified by 16S rDNA sequencing.

Most of the isolates belonged to LAB species (85; 57%), while numerous belonged to the *Enterobacteriaceae* family (63; 43%). Most of the LAB isolates (64) were identified as *Lactiplantibacillus* spp. (basionym *Lactobacillus* spp.), belonging to the *plantarum*, *paraplantarum*, *pentosus*, or *argentoratensis* species. The highest number of *Lactiplantibacillus* spp. isolates were isolated from olives (59), while only five were isolated from brines. Other olive LAB isolates were identified as *Leuconostoc mesenteroides*/*pseudomesenteroides* (19), *Lactococcus lactis* (1), and *Enterococcus* spp. (1). As expected, due to phylogenetic similarities, 16S rRNA gene sequencing did not clearly discriminate *L. plantarum* from *L. paraplantarum*, *L. pentosus*, or *L. argentoratensis* [93]; *E. faecium* from *E. faecalis* or other enterococcal species [94]; or *L. mesenteroides* from *L. pseudomesenteroides* [95], but further discrimination was out of the scope of the present study.

As presumed, 75% of the dominant LAB isolates were lactobacilli, isolated from day 4 to day 90, while only 22% of them were *Leuconostoc* spp. isolated from olives, either untreated or up to day 30 of fermentation. Several research articles and reviews in the last four decades describe the prevalence of LAB in the microbial ecosystem of Greek-style olives, and their parallel impact with fungi on sensorial properties and shelf life [96], as well as their human health-related traits [97,98,99]. *L. plantarum* and *L. pentosus* have been previously reported as the most frequently isolated LAB species from fermented olives and/or brines [24,25,97,100,101]. De Angelis et al. [26] also reported that *L. plantarum* and *L. pentosus* dominated the total and metabolically active microbiota of the control (without starter cultures) olives and brines at the end of the fermentation. Moreover, Heperkan [102] reported in a review that the most relevant LAB species to be used as starter cultures, in single or mixed combinations, for naturally fermented (Greek-style) as well as Spanish-style olives are *L. plantarum* and *L. pentosus*. The fact that *L. plantarum* and *L. pentosus* are the dominant LAB species of naturally fermented green and black olives has also been verified by multiple studies using amplicon-based sequencing analysis, as mentioned above [24,25,26,27,28,29,30,31,32,33]. They are considered to play a significant role in shaping the olives’ sensory profile, not only by exhibiting β-glucosidase and esterase activity and therefore being involved in the debittering of olives, but also because of their facultative heterofermentative metabolism [99]. Heterofermentative LABs, such as *Leuconostoc* spp., are also commonly detected in fermented olives [101,103,104]. On the other hand, salt-resistant enterococcal species, although not predominant during olive fermentation, are members of LAB other than lactobacilli and *Leuconostoc* spp., present at lower concentrations along with streptococci and pediococci [105]. Although the role of enterococci during the fermentation of table olives is not clear, the isolation of *Enterococcus* spp. from olive brines has been reported by several authors, as reviewed by De Castro et al. [106], and several strains have been used as starters along with other LABs and/or yeasts [106].

Concerning the *Enterobacteriaceae* isolates, most of them were identified as *Mangrovibacter* spp. (totally 39, 6 from olives and 33 from brines), 22 (all from olives) as *Enterobacter*/*Raoultella*/*Klebsiella* spp., one (from olives) as *Kluyvera intermedia*, and one (from brine) as *Pseudocitrobacter* spp. Isolates from this family were found in olives up to day 30 and in brines up to day 15, which is consistent with the respective microbial counts being zeroed between day 15 (brine) and day 30 (olives). Although the fermented olive microbiome is dominated by LABs and fungi, members of *Enterobacteriaceae* have been detected at the beginning of the process, possibly generating off-odors and -flavors [25,26,104,106]. This group is eliminated during fermentation and is not detected at the end of the process [32,107]. The genus *Mangrovibater* was described in 2010 [108], and thereafter, multiple reports suggest its abundance and wide distribution across diverse habitats [109], including high-salt sludges and geothermal lakes, root systems, foliage and fruits of plants, where it can exist in an endophytic form, and in several organisms throughout the animal kingdom, as a symbiotic bacterium [109]. The high abundance and wide distribution of *Mangrovibacter* across diverse habitats indicates its multi-functional role in various ecosystems. In plants, like other endophytic microorganisms, it is possible that it contributes to the growth, by enhancing nutrient absorption, bolstering stress tolerance, and fortifying resistance against diseases; therefore, it may contribute to the enhancement of crop yields [110]. The presence of *Mangrovibacter* has been recently reported in directly brined green table olives [111], as well as in olive mill wastewater sludge [112]. Moreover, Mennane et al. [113] also reported the occurrence of *Klebsiella* spp. and *Enterobacter sakazakii* in Moroccan fermented olives. Several authors demonstrated that the olive fermentation environment supports the growth/survival of pathogen microorganisms, such as *Enterobacter cloacae*, an opportunistic pathogen for humans, recovered in spontaneously fermented table olives [114]. Interestingly, it is reported that there was a parallel (positive) presence of the genera *Enterobacter*, *Raoultella*, *Klebsiella*, *Mangrovibacter*, *Escherichia*, *Shigella*, *Clostridum sensu stricto*, and *Atlantibacter* in spoiled fermented olives, although these genera were associated with low levels of salt, i.e., 1.2–1.4% *w*/*v* [52].

### 3.4. Rep-PCR Fingerprinting and Identification of Fungal Isolates

A total of 178 fungal colonies were isolated (YGC agar plates, 25 °C) from olives (127) and brines (51) (Table 2 and Appendix A). Based on the rep-PCR analysis, fungal isolates were clustered in 45 groups (Appendix A). Representative isolates from each group were identified by ITS rDNA sequencing. However, there was no rep-PCR product for five isolates, namely F128, F180, F194, F200, and F256; therefore, since they could not be clustered, they were also identified by ITS rDNA sequencing.

Among the 178 isolated fungi, 136 isolates were yeasts collected from olives (96) and brines (40). Most of the yeasts belonged to the genera *Candida* (35.3%) and *Nakazawaea* (31.6%), which were isolated from untreated olives as well as from both fermented olives (65 isolates) and brines (26 isolates) at all time points of fermentation. Concerning the species, the *Candida* isolates were identified as *C. boidinii* (24), *C. diddensiae* (22), *C. adriatica* (1), and *C. neodendra* (1); all *Nakazawaea* isolates (43) were identified as *N. molendinolei* (basionym: *Candida molendinolei*). *Saccharomyces cerevisiae*/*boulardii*/*paradoxus* (21 isolates) and *Pichia kluyveri*/*fermentans* (16 isolates) were also isolated. *S. cerevisiae*, *S. boulardii*, and *S. paradoxus* are reported as sibling species with close phylogenetic relatedness; therefore, the sequencing of the ITS region is not sufficient for their identification, and complementary methods, such as restriction digests of the ITS PCR products, are required [115,116]. However, concerning *S. cerevisiae*, the most emblematic and industrially relevant yeast species, it has been recently proposed that the strains in olives originate from a hybridization between *S. cerevisiae* and *S. paradoxus* [117]. It is worth noting that *S. cerevisiae* has a long list of taxonomical synonyms, including *S. boulardii* and *S. diastaticus*. [118]. Furthermore, for *P. kluyveri* and *P. fermentans*, distinguishing methods, such as the PCR-RFLP analysis of the 5.8 ITS rRNA region and sequence information for the D1/D2 domains of the 26S rRNA gene, are necessary [119]. However, further discrimination was beyond the scope of the present study. The rest of the eight isolates were identified as *Barnettozyma californica* (6), *Rhodotorula glutinis*/*graminis* (1), and *Cyberlindnera fabianii* (1). *Candida*, *Nakazawaea*, and *Saccharomyces* were the only genera isolated at the end of fermentation. This fact seems to be in accordance with studies reporting that a broader range of yeast species can be detected in the beginning rather than the end of the fermentation [120,121].

The isolation of *Candida* spp., *Saccharomyces* spp., *Pichia* spp., and *Rhodotorula* spp. was expected because, staring in 1965, early studies of olive fermentations have mentioned the presence of most of them, as reported by Arroyo-López et al. [122]. Arroyo-López et al. [123] included the detection of the isolated genera/species of the present study in their review of multiple studies.

During the last few years, the role of yeasts has been re-evaluated and appreciated for both green-treated and black naturally fermented olives [124]. Yeasts can play an important role during olive fermentation, i.e., they exhibit desirable biochemical activities (lipase, esterase, β-glucosidase, catalase, production of killer factors, etc.), offering essential technological benefits to the final product [59]. The production of volatile compounds, originating from the yeasts’ metabolism, is the most important yeast property contributing to the final organoleptic character of fermented olives [96,125,126]. Several yeast strains isolated from Portuguese brined olives exhibited the absence of pectinolytic activity, a positive catalase response, high osmotolerance, the ability to take up oleuropein and lactic acid, and the capacity to produce B-complex vitamins [62]. There are many references concerning the biotechnological potential of the yeast species isolated during the present study, as mentioned below. A strain of *N. molendinolei* was included in a study evaluating yeast starters for the production of industrial-scale Taggiasca black table olives [127]. *C. boidinii* is widespread in nature and has been isolated from diverse substrates related to human activity, including olive manufacturing [128]. It is a xylose-consuming and methylotrophic yeast and, therefore, a species presenting clear biotechnological potential. When involved in olive processing, it exhibits different multifunctional features, such as lipase activity, biofilm formation on fruit epidermis, and co-aggregation with LAB species, such as *L. pentosus*, as reviewed by Camiolo et al. [128]. The concomitant importance of *S. cerevisiae* lies in its unique biological characteristics, e.g., fermentation capacity and production of alcohol and CO_2_, which contribute to its various biotechnological applications [129]. *S. cerevisiae* exhibiting glucosidase activity may be involved in the hydrolysis of oleuropein present in olive oil and may therefore contribute to the fading of the bitter taste of newly produced olive oil during storage [130]. *P. kluyveri* isolated from fermented olives improved the volatilome of Hojiblanca naturally fermented green olives [31]. *P. fermentans* is another “ester aromatic yeast” that produces polysaccharides [131]. *C. fabianii* is a species widely distributed in the environment and has been used in biotechnological procedures, as reviewed by Colautti et al. [132]. However, to our knowledge, it has not been isolated from olives before. Finally, Randazzo et al. [25] highlighted the interaction between LAB and yeasts as mentioned above, since some yeast species seem not only to contribute to the aroma compound production but also stimulate the growth of LAB strains [123,133,134,135,136].

However, yeasts have been blamed for spoiling olives, as they exhibit pectolytic and xylanolytic activities and may produce off-flavors and -odors, brine-clouding, gas-pocket formation, and package bulging [124]. *Rhodotorula* species have been detected in table olive fermentation and were reported to cause the softening of fruits exhibiting protease, xylanase, and pectinase activity [137]. *C. fabianii* has been recently described as a contaminant of food and fermentation products [132], and it has also been reported as an opportunistic pathogen isolated from clinical specimens [138]. Moreover, yeasts may be involved in the production of undesired acidity in olive oil due to the hydrolysis of triacylglycerols [139,140,141]. Nevertheless, olive oil’s chemical composition is important, and the improvement of its flavor after a period of storage is assigned to enzyme activities of not only plants but also microbes, including fungal origin, during its extraction or sedimentation phase [130,142]. For example, yeasts including *C. adriatica* and *C. diddensiae*, originating from the fruit, can damage the quality of the olive oil and therefore are considered harmful; however, the water and polyphenol content of olive oil can modulate their lipolytic activity [141].

Concerning yeast-like or non-yeast fungi, 42 were isolated (Table 2 and Appendix A) from olives (31 isolates) and brines (11 isolates). Taking into consideration the new nomenclature code adopting the “one fungus, one name” principle [143], most of the isolates (31) were identified as *Geotrichum candidum* (synonym: *Galactomyces candidus*)/*Geotrichum australiense* (basionym: *Dipodascus australiensis*)/*Geotrichum galactomycetum* (basionym: *Galactomyces geotrichum*) [144]. Species of *Geotrichum* were isolated from olives until the end and from brines until day 30 of fermentation. Additionally, four isolates (isolated from untreated olives and from olives at day 0 and 4) were identified as *Aureobasidium pullulans*, two (olives, day 15) as *Filobasidium magnum* (basionym: *Cryptococcus laurentii* var. *magnus*), one (brine, day 15) as *Cladosporium halotolerans*, one (olives, day 15) as *Cladosporium tenellum*/*herbarum*/*ramnotenellum*/*iridis*, one (olives, day 0 of fermentation) as *Lasiodiplodia macrospora*, one (brine, day 4) as *Fusarium solani*/*hoffmannii*/*ambrosium*/*ensiforme*, and one (brine, day 15) as *Aspergillus amstelodami*/*cristatus*/*chevalieri*/*montevidensis*.

*Geotrichum candidum* is a ubiquitous filamentous yeast-like fungus mostly known for its use as a starter in the dairy and brewing industry, but it is also important for agrifood- and bio-industries, comprising strains producing antimicrobial compounds and being capable of bioremediation [145]. It has been isolated from table olives [122] and from alpeorujo, a residue of olive oil production [146]. Additionally, it has been isolated from olive mill wastewater and was used for its treatment, i.e., to control decolorization and biodegradation [147]. The cactophilic *D. australiensis* (current name: *G. australiense*) was one of the species that dominated the fungal microbiome of olive mill wastewater sludge, while *Galactomyces* sp. seemed to be strongly involved in phenol biodegradation and to influence the composting process [148]. The detection of *A. pullulans* has been reported in green Konservolia [120], black Kalamata [121], and green and black Taggiasca [126] naturally fermented olive varieties. This ubiquitous and widespread black yeast-like species is particularly known for its biotechnological potential, as it produces pullulan (poly-α-1,6-maltotriose), a biodegradable extracellular polysaccharide [149]. *F. magnum* has been isolated from rhizosphere soil [150] and, recently, from fermented natural black olives [32]. *Cladosporium* spp. has been isolated from Portuguese olive varieties [151], black Greek-style fermented Konservolia olives [29], Spanish naturally fermented olives [152], and it has been detected in Italian Taggiasca olives by metataxonomic analysis [126]. To our knowledge *C. halotolerans*, *C. tenellum*, *C. herbarum*, *C. ramnotenellum*, and *C. iridis* have not been isolated from olives before. Different species of the soil-borne *Fusarium* genus, i.e., *F. oxysporum*, *F. solani*, *F. graminearum*, and *F. eupionnates*, have been isolated from multiple olive parts, either healthy or exhibiting different pathogenic symptoms [153,154,155]. The genus *Lasiodiplodia* is an important plant pathogen with multiple hosts and may cause fruit and root rot, die-back of branches, as well as stem canker [156]; however, to our knowledge, *L. macrospora* has not been isolated from olives before. Finally, *Aspergillus* spp. has been isolated from the fruits of Portuguese olive varieties [151], while this genus was one of the most abundant fungal genera in Cypriot spontaneously fermented olives, as revealed through metabarcoding sequencing [30].

### 3.5. Metataxonomic Analysis

#### 3.5.1. Bacterial Microbiota Composition

A total of 19 bacterial genera were detected in the samples (Appendix A). Brine samples presented higher α-diversity (ANOVA, *p*-value < 0.05) when compared with olive ones (Appendix A). The α-diversity showed lower Simpson’s and inverted Simpson’s indexes of olives fermented after 30 and 90 days when compared to untreated olives (ANOVA, *p*-value < 0.05) (Appendix A).

Brines showed the highest abundance of *Enterobacteriaceae*, *Shigella*, *Lactiplantibacillus*, and *Leuconostoc* (Figure 1 and Appendix A). *Coleofasciculaceae* was the most abundant ASV in olives (>60%), *Lactiplantibacillus* was more abundant in olives after day 7 (7.2%), while *Leuconostoc* was the most abundant after days 30 and 90 (5.2 and 5.5%, respectively) (Figure 1 and Appendix A). Untreated olives showed that the second taxon in terms of abundance was *Enterobacteriaceae* (8.6%) (Figure 1 and Appendix A). *Coleofasciculaceae* and *Novosphingobium* spp. were more abundant in olives at day 30 (LDA = 5.81) and at day 90 (LDA = 5.18), respectively (LEfSe, *p*-value < 0.01).

The PCoA based on Bray–Curtis’s distance matrix of fermented olives showed differences in terms of bacterial taxa when compared to brines (Figure 2).

Several studies using metagenomics analysis have reported the detection of the above-mentioned bacterial genera, such as *Lactiplantibacillus* in Nyons variety [32], Manzanilla variety [111], Cypriot, Picual, and Kalamata varieties [30]; *Leuconostoc* in Konservolia variety [29], Kalamata variety [33], Nyons variety [32], Cypriot, Picual, and Kalamata varieties [30]; *Shigella* in Manzanilla spoiled Spanish-style fermented olives [52]; and *Novosphingobium* in untreated Aloreña de Málaga olives [28] and Kalamata variety [33], as well as the detection of the families *Lactobacillaceae* in Konservolia variety [29] and Kalamata variety [33], and *Enterobacteriaceae* in Aloreña de Málaga olives [28], Nocellara Etnea variety [25], Konservolia variety [29], Kalamata variety [33], Nyons variety [32], and Hojiblanca variety [31]. However, to our knowledge, the *Coleofasciculaceae* family, which corresponds to terrestrial cyanobacteria playing a significant role as pioneers of biological soil crusts [157], has not been detected in olives using metataxonomic analysis, although it is possible that its presence has not been reported because of its very low abundance. The presence of *Coleofasciculaceae* in olives could be attributed to contamination from the soil; however, it is possible that this family is a member of the olive microbiome, as its presence is also verified in leaves by metataxonomic analysis during the present study. Moreover, cyanobacteria species have already been isolated from plants’ phyllosphere, e.g., Acanthaceae and Poaceae [158]. Today, cyanobacteria play an important role in global oxygen, carbon, and nitrogen cycles [159]; thus, it is particularly intriguing to investigate their presence in olive fruits.

The results of the metataxonomic analysis are in accordance with the results of the classical microbiological analysis, as a high proportion of the isolated bacterial strains belonged to the *Enterobacteriaceae* family (63 isolates) and the *Lactiplantibacillus* and *Leuconostoc* genera (64 and 19 isolates, respectively). *Enterobacteriaceae* were eliminated due to pH decrease and the presence of phenolic compounds during the fermentation, as revealed by the classical microbiological analysis. They were, however, detected by metataxonomic analysis until the end of the process in both olive and brine samples, as this analysis enables the detection of both live and dead microbial cells. Moreover, many of the families and genera corresponding to the bacterial ASVs detected were not isolated from either olives or brines, e.g., *Coleofasciculaceae* and *Novosphingobium*, while all isolated genera were detected by the metataxonomic analysis. This was expected, as metataxonomics, using evolved sequencing technologies, can detect a broader range of microbial genera, including those not easily cultured using the growth media used in this work, e.g., members of the *Coleofasciculaceae* family [160]. Moreover, opportunities to guide the isolation and cultivation of microbes of interest are provided by the growing abundance of metagenomic sequence information from a wide range of environments [161].

#### 3.5.2. Mycobiota Composition

A total of twenty-six (26) fungal taxa at the lowest taxonomic resolution were detected (Appendix A), whereas those detected but not specifically identified were categorized collectively as ‘Others’ in Figure 3. Mycobiota (Figure 3) of olives and brines did not demonstrate differences in terms of α-diversity (Appendix A). *Saccharomyces* was the most abundant ASV in olives and brines (>70%), followed by *Nakazawaea* (>9%) in both fermented olives and brines (Figure 3). The second-most abundant taxa of untreated olives and olives at day zero were *Cyberlindnera* (5.3%) and *Cytospora* (13.4%), respectively (Figure 3). PCoA did not show separation between sample groups (Appendix A). *Cyberlindnera* was associated with olives at day 0 (LEfSe *p*-value < 0.05, LDA = 4.44).

Several studies using metataxonomic analysis have also reported detection of the above-mentioned fungal genera *Saccharomyces* [32] and *Nakazawaea* [111]. Additionally, *Saccharomyces* was one of the dominant genera in fermented olives, such as Aloreña de Málaga variety [27], Kalamata variety, [33], Manzanilla variety [111], Picual and Kalamata varieties [30], and Hojiblanca and Manzanilla varieties [31]; however, a low occurrence (<1%) of *Saccharomyces* was observed in the yeast consortium in fermented Greek table olives of Halkidikis and Konservolia varieties [29]. Nevertheless, to our knowledge, the presence of the genera *Cyberlindnera* and *Cytospora* in olives has not been so far reported, possibly due to its low abundance as reported above.

The results of the metataxonomic analysis are in accordance with the results of the classical microbiological analysis, as a high proportion of the isolated fungal strains belonged to the *Saccharomyces* and *Nakazawaea* genera (21 and 43 isolates, respectively), and one isolate was identified as *Cyberlindnera* sp. However, most of the genera and families corresponding to the fungal ASVs detected were not isolated either from olives or brines, e.g., the genus *Cytospora,* which was detected in olives at the beginning of fermentation. This filamentous fungal genus commonly inhabits woody plants, includes several important canker-causing species, and has been reported to affect olive trees [162]. On the other hand, all genera isolated from olive and brine samples were detected by the metataxonomic analysis, with the exception of the genus *Lasiodiplodia* (one isolate). This result was not surprising either, as one of the challenges of the sequencing process for metataxonomic analysis may miss low-abundance microorganisms [160]. The isolation of microorganisms remains critical to directly study them and confirm their metabolic and physiological functions, while cultivated isolates could be useful in a range of applications, such as new probiotics, biocontrol agents, and agents for industrial processes [161]. The high abundance of a fungal species in a single sample, i.e., *Cytospora* in sample CO0 (41.3% abundance), indicates that the sampling process influences the results of metataxonomic analysis [163].

### 3.6. Sensory Evaluation of Fermented Tsounati Table Olives

The sensory profile (original scores) of Tsounati olives after 90 days of fermentation is presented in a spider graph (Figure 4).

The evaluation scores of the panel members were similar concerning both gustatory (salty, bitter, acid) and kinesthetic (hardness, fibrousness, crunchiness) sensations (Appendix A). Average score values were 5.6 ± 0.1 (salty), 5.8 ± 0.4 (bitter), 4.6 ± 0.2 (acid), 6.3 ± 0.1 (hardness), 6.4 ± 0.2 (fibrousness), and 7.0 ± 0.1 (crunchiness). On the contrary, the commercial samples included in the analysis were perceived to be saltier and more acidic, however, less bitter, indicating higher salt concentration in the brine, longer period of fermentation, and possible addition of acid, respectively. Concerning smell and color, all samples had similar score values (7.04 ± 0.4, 6.06 ± 0.3 respectively). Abnormal fermentation was not indicated for any of the samples. The panelists not only did not perceive spoilage odors, such as those associated with zapatera spoilage, butyric, sulfidic, or putrid fermentation [52], but they also highlighted the mild and fruitful taste of Tsounati fermented olives.

### 3.7. Overall Discussion

The exploration of microbial diversity in fermented foods has been significantly advanced by the application of high-throughput sequencing (HTS) combined with omics approaches. However, culture-dependent analysis remains essential for estimating viable counts of various microbial groups and for isolating microorganisms for potential future applications. This necessity arises from limitations of culture-independent methods, such as the inclusion of DNA from dead or metabolically inactive cells, the insufficient detection of microbial populations present at low abundance, and variability in DNA quality extracted from microbial communities, all of which can affect sequencing success [164].

The microbiome of olives is closely linked to specific characteristics, including structural, chemical, genetic, and physiological traits. The uniqueness and appeal of traditionally fermented table olives stem from the complex and dynamic fermentation process, driven by a consortium of interacting microorganisms that collectively shape the final product. Furthermore, the abundance, prevalence, and succession of microbial populations during fermentation are influenced by various internal and external factors that affect the process [165].

Abiotic factors, such as weather conditions, including temperature, rainfall, and humidity, along with cultivation practices (traditional or novel), fruit maturation stage, microbial endophytic or epiphytic status, soil characteristics, salinity, and pollutant levels can all influence olive trees and negatively impact both yield and quality, as well as the microbiome of olive fruits [165,166,167].

Biotic stresses, such as olive tree diseases caused by arthropods, bacteria, fungi, nematodes, and viruses, interact with and influence microbial survival and dominance [166]. Additionally, the composition of the endophytic microbial community may vary according to plant age, altitude, and geographic location [167]. Recently, a study investigating the microbial terroir of fresh olive samples from 38 Greek olive varieties, including Tsounati, collected across geographically diverse regions of Greece, was conducted employing metataxonomics [168]. This study revealed a notable association between geographical origin, tree age, altitude, and microbial composition, underscoring their potential role in shaping the fermentation process.

The impact of olive cultivar genotype on the bacterial communities present in olive fruits and leaves has been studied, suggesting that genotype may have a stronger influence than soil and climate conditions, as reported in the reviews by Cardoni and Mercado-Blanco [166] and Melloni and Cardoso [167]. The microbial community of Tsounati fermented olives analyzed in this study, using both culture-dependent and -independent methods, shared several bacterial and fungal taxa commonly found in other olive varieties, including *Lactobacillus plantarum*, *L. pentosus*, *Leuconostoc* spp., and *Enterobacteriaceae*, and fungi such as *Candida*, *Saccharomyces*, *Pichia*, *Rhodotorula*, and *Geotrichum candidum*. However, several taxa not previously detected in table olives, such as *Coleofasciculaceae*, *Cyberlindnera*, *Cytospora*, and *Candida fabianii*, were identified, possibly due to the factors mentioned above, although the possibility remains that these taxa were present but undetected in earlier studies.

The microbial consortium plays a crucial role during olive fermentation, exhibiting both desirable and undesirable biochemical activities while providing essential technological benefits to the final product, as previously mentioned. However, this consortium is itself directly influenced by the numerous factors discussed above. This multifactorial interaction creates a complex ecosystem within table olives. The phenolic compounds and triterpenic acids present in Tsounati table olives may not only confer human health benefits but also affect the microbial consortium, which in turn directly shapes the distinctive sensory characteristics of Tsounati olives.

## 4. Conclusions

In the present study, naturally fermented olives of the Greek Tsounati variety from Monemvasia area were produced. The high abundance of *Lactiplantibacillus* and *Leuconostoc*, as well as *Saccharomyces* and *Nakazawaea* until the end of the fermentation, determined by classical microbiological and metataxonomics analyses, showed that both olives and brines were dominated by LAB and fungi known for their desirable biochemical activities and beneficial impact on the organoleptic character of the fermented olives. However, the detection of *Enterobacteriaceae*, as well as *Cyberlindnera* and *Cytospora*, suggests the occurrence of undesirable bacteria and fungi, which may have spoilage potential or even pathogenic traits. Moreover, members of the aforementioned LAB and fungal microbiota were isolated and identified, confirming the existence of these microorganisms until the end of fermentation and pointing out the disappearance of *Enterobacteriaceae* after 30 days. The presence of the filamentous fungal species *Geotrichum candidum* in olives until the end of fermentation reinforces the ambiguous role of fungi. TPC was lower compared to that of other olive varieties previously reported, possibly due to irrigation, but it remained largely unaffected during fermentation. Additionally, two of the most abundant and important bioactive compounds of olive fruits, the triterpenic acids maslinic and oleanolic, were monitored and quantified by HPLC. Their consistent presence showed that natural fermentation did not influence their concentration and suggested multiple beneficial health effects for consumers. Finally, sensory evaluation revealed the mild and fruitful taste of Tsounati fermented olives and strengthened their potential for table olive production. The results show the importance of the Tsounati variety as a promising Greek olive variety for future applications.

## Figures and Tables

**Figure 1 foods-14-02568-f001:**
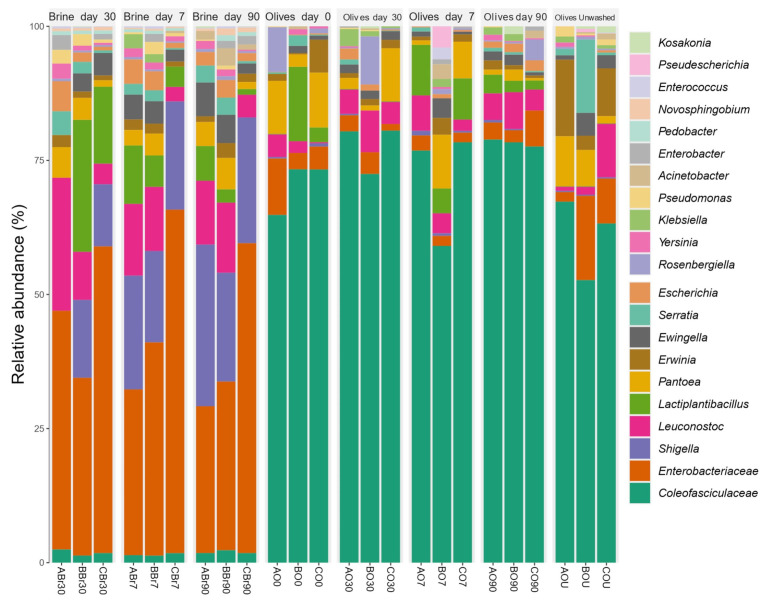
Bacterial taxa relative abundances of olives and brines based on the metataxonomic analysis.

**Figure 2 foods-14-02568-f002:**
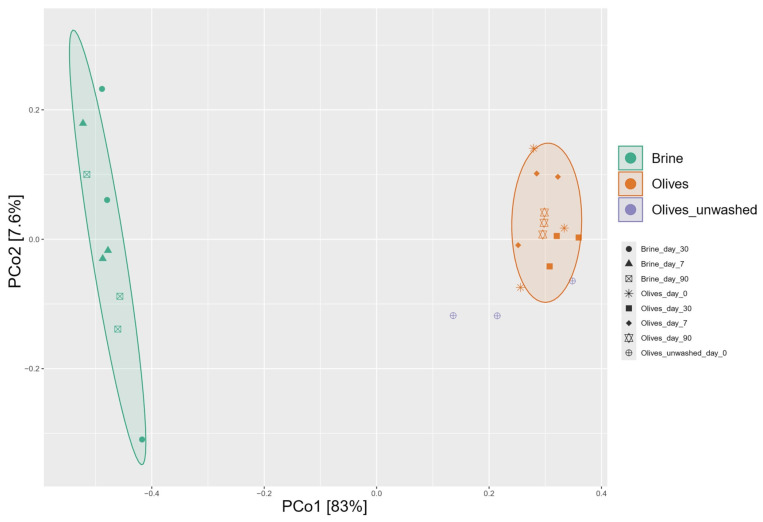
PCoA analysis about bacterial β-diversity (Bray–Curtis’s distance) of olives and brines based on the metataxonomic analysis.

**Figure 3 foods-14-02568-f003:**
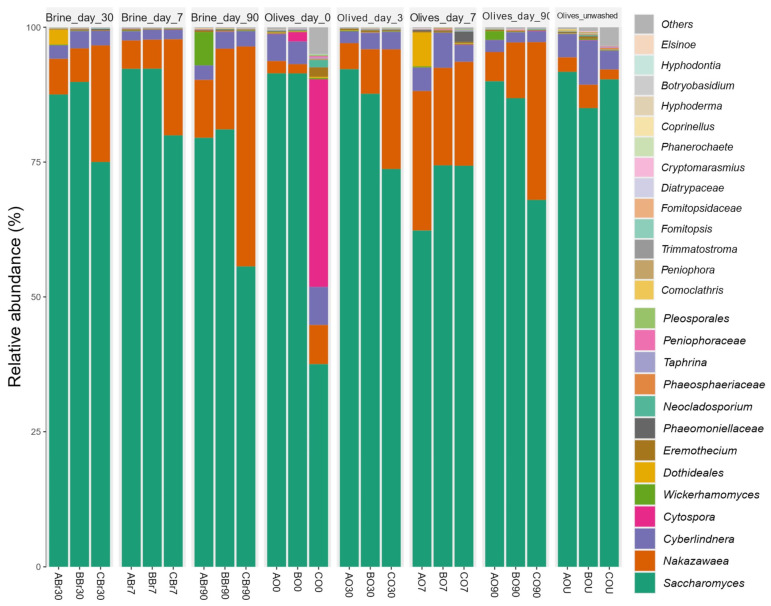
Fungal taxa relative abundances of olives and brines based on the metataxonomic analysis.

**Figure 4 foods-14-02568-f004:**
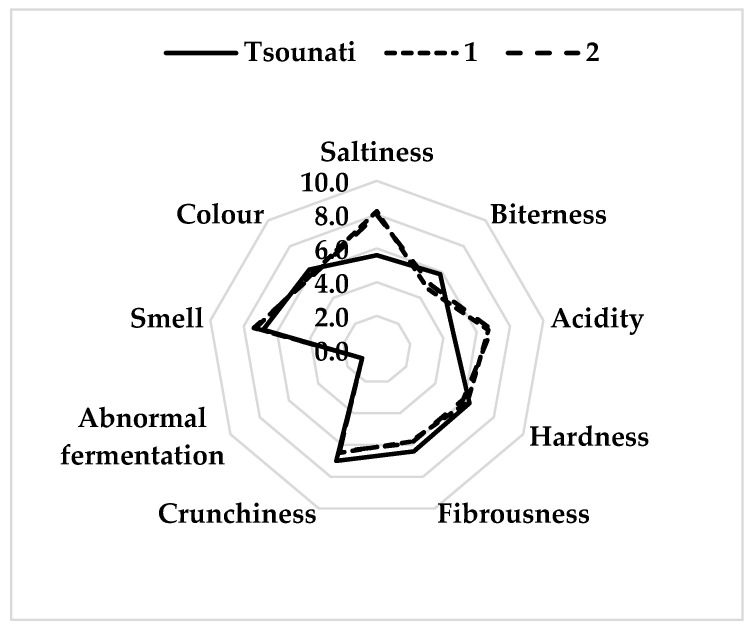
The spider graph shows the sensory profiles (original scores) of Tsounati olives after fermentation of 90 days based on the sensory evaluation. Fermentations A, B, C; 1, 2: Commercial samples of green Konservolia variety fermented table olives.

**Table 1 foods-14-02568-t001:** Number of identified bacterial isolates (isolated from MRS agar plates incubated at 30 °C) from Tsounati olive and brine samples.

Identification	OU	O0	O4	O7	O15	O30	O60	O90	Br4	Br7	Br15	Br30	Br60	Br90	Nr
Lactic Acid Bacteria															85
*Lactiplantibacillus* spp.	0	0	5	8	9	3	6	28	0	1	2	0	0	2	64
*Leuconostoc* spp.	3	1	6	5	2	2	0	0	0	0	0	0	0	0	19
*Lactococcus lactis*	0	0	0	1	0	0	0	0	0	0	0	0	0	0	1
*Enterococcus* spp.	0	0	0	0	1	0	0	0	0	0	0	0	0	0	1
*Enterobacteriaceae*															63
*Mangrovibacter* spp.	2	0	2	0	1	1	0	0	14	11	8	0	0	0	39
*Enterobacter*/*Raoultella*/*Klebsiella* spp.	7	14	0	1	0	0	0	0	0	0	0	0	0	0	22
*Kluyvera intermedia*	1	0	0	0	0	0	0	0	0	0	0	0	0	0	1
*Pseudocitrobacter* spp.	0	0	0	0	0	0	0	0	1	0	0	0	0	0	1
Number of isolates per sample/ Total number	13	15	13	15	13	6	6	28	15	12	10	0	0	2	148

O: olives (fermented), U: untreated olives, Br: brine, Nr: number.

**Table 2 foods-14-02568-t002:** The number of identified fungal isolates (isolated from YGC agar plates incubated at 25 °C) from Tsounati olive and brine samples.

Identification	OU	O0	O4	O7	O15	O30	O60	O90	Br4	Br7	Br15	Br30	Br60	Br90	Nr
Yeasts															136
*Candida* spp.	2	4	2	5	6	4	8	4	2	3	3	2	0	3	48
*Nakazawaea molendinolei*	1	0	2	4	4	4	9	6	3	3	2	2	0	3	43
*Saccharomyces cerevisiae*/*boulardii*/*paradoxus*	0	0	0	1	3	3	5	3	1	3	0	2	0	0	21
*Pichia kluyveri*/*fermentans*	1	2	3	2	1	1	1	0	2	2	1	0	0	0	16
*Barnettozyma* *californica*	0	2	2	0	0	0	0	0	1	0	1	0	0	0	6
*Rhodotorula glutinis*/*graminis*	0	0	0	0	0	0	0	0	1	0	0	0	0	0	1
*Cyberlindnera* *fabianii*	1	0	0	0	0	0	0	0	0	0	0	0	0	0	1
Yeast-like or non-yeast Fungi															42
*Geotrichum candidum*/*australiense*/*galactomycetum*	2	1	3	3	3	3	6	2	0	3	2	3	0	0	31
*Aureobasidium pullulans*	2	1	1	0	0	0	0	0	0	0	0	0	0	0	4
*Filobasidium magnum*	0	0	0	0	2	0	0	0	0	0	0	0	0	0	2
*Cladosporium halotolerans*	0	0	0	0	0	0	0	0	0	0	1	0	0	0	1
*Cladosporium tenellum*/*herbarum*/*ramnotenellum*/*iridis*	0	0	0	0	1	0	0	0	0	0	0	0	0	0	1
*Aspergillus amstelodami*/*cristatus*/*chevalieri*/*montevidensis*	0	0	0	0	0	0	0	0	0	0	1	0	0	0	1
*Fusarium solani*/*hoffmannii*/*ambrosium*/*ensiforme*	0	0	0	0	0	0	0	0	1	0	0	0	0	0	1
*Lasiodiplodia macrospora*	0	1	0	0	0	0	0	0	0	0	0	0	0	0	1
Number of isolates per sample/Total number	9	11	13	15	20	15	29	15	11	14	11	9	0	6	178

O: olives (fermented), U: untreated olives, Br: brine, Nr: number.

## Data Availability

The original contributions presented in the study are included in the article/Appendix A, further inquiries can be directed to the corresponding author.

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
