# Peer review of "The Natural Fermentation of Greek Tsounati Olives: Microbiome Analysis"

_foods, 2025, doi:10.3390/foods14152568_

Round 1
Reviewer 1 Report
Comments and Suggestions for Authors
The authors of the manuscript analyzed the microbial community of Greek Tsounati olives after three months of fermentation, and evaluated the value of the fermented products by testing the total polyphenols and triterpenoids. The manuscript has a perfect experimental design, sufficient data, accurate analysis, and smooth logical expression, and has high academic value. On this basis, the author needs to modify some deficiencies in the article to make the article closer to perfection.
- Could the authors briefly describe the experimental methods used in the abstract?
- The introduction contains a lot of content that is not directly related to Tsounati olives (such as a basic overview of olives, etc.). It is recommended to simplify it.
- In the HPLC part of the method, although existing literature is cited, it is still recommended to indicate the equipment model and column type.
- The relationship between the salt content of edible olives and salt intake is discussed in 3.1.1 of the results section, but the author failed to handle the statement here well, making the discussion of salt intake abrupt and failing to clearly state the purpose of the discussion. It is recommended to add connecting text.
- Will lab fermentation differ from fermentation in production scenarios, for example, resulting in different polyphenol content due to changes in oxygen content? Or will changes in fermentation tank volume produce unpredictable chain reactions?
- During the production process, the initial fermentation flora may be different due to different origins and process equipment. Can the author's research cover this potential variable factor?
- The author's description of the experimental results is scattered in each chapter, and the summary in the conclusion is too brief. It is recommended that the author add a discussion chapter to discuss the experimental data and results after refining them, connect the data from each part, and then make a final summary in the conclusion.
- Some references are too old. If possible, it is recommended to replace them with more recent ones.
Reviewer 2 Report
Comments and Suggestions for Authors
This study presents a thorough investigation into the microbial diversity and fermentation dynamics of Tsounati olives, a Greek olive variety traditionally cultivated for oil production and rarely explored for table olive processing. Over a 90-day natural fermentation period, the research aims to characterize the bacterial and fungal communities involved, using both culture-dependent techniques and metataxonomic sequencing. The study not only identifies dominant microorganisms but also tracks key physicochemical changes throughout fermentation. In addition, a sensory evaluation was conducted to assess the commercial potential of Tsounati olives as a table product. The originality and relevance of this research lie in its focus on the Tsounati variety, which has not been previously studied in such depth. While most olive fermentation studies concentrate on popular varieties such as Kalamata or Konservolia, this is the first to offer a comprehensive microbial and biochemical profile of Tsounati olives. The unexpected detection of Coleofasciculaceae (cyanobacteria) and the yeast Cyberlindnera expands current understanding of olive-associated microbiomes. These findings suggest that Tsounati olives may have untapped potential for diversification in Greek table olive production. Compared to earlier studies, particularly those on the Kalamata variety, this work stands out for its detailed metataxonomic analysis. However, it lacks comparative data on fermentation efficiency, such as acidification rates, which could have strengthened the conclusions. The study’s strengths include its multi-method approach, combining culturing techniques, rep-PCR fingerprinting, and high-throughput Illumina sequencing for robust microbial identification. It also provides a comprehensive chemical profile by monitoring changes in sugars, organic acids, polyphenols, and triterpenic acids. These data are crucial for understanding the biochemical landscape during fermentation.
However, the study has some minor limitations, that do not affect the overall quality. A noteworthy concern is the final pH of 4.73, which exceeds the recommended safety threshold of 4.3. Although the authors suggest that high levels of phenolic compounds may have inhibited lactic acid bacteria and thus limited acidification, the potential risks related to spoilage and food safety warrant further discussion.
The conclusions of the study are generally consistent with the data presented. In summary, this work represents a significant contribution to the field of olive fermentation, particularly by highlighting the potential of the Tsounati variety for table olive production.
Reviewer 3 Report
Comments and Suggestions for Authors
The manuscript entitled “The Natural Fermentation of Greek Tsounati Olives: Microbiome Analysis” is an interesting topic. However, there are still several parts that need to be revised:
- The introduction introduces the background of olive varieties and fermentation process in detail, but lacks clear research objectives and assumptions. It is suggested that the specific objectives and scientific issues of this study should be put forward more clearly at the end of the introduction, so that readers can better understand the motivation and importance of the study.
- In the part of materials and methods, some key steps are not described in detail. For example, the adjustment frequency and specific method of salt concentration during fermentation are not clearly stated, which may affect the repeatability of the experiment. It is suggested to supplement these details to ensure that other researchers can reproduce the experiment.
- Traditional microbiological methods and macrotaxonomy analysis were used in the study, but the limitations of the two methods and their effects on the results were not discussed. For example, traditional culture methods may omit uncultured microorganisms, and macrotaxonomy may be affected by DNA extraction and sequencing deviations. It is suggested that these contents be supplemented in the discussion section.
- The statistical analysis mentioned in the results (such as ANOVA and Tukey test) did not provide specific significance level (such as P value) or effect quantity. It is suggested that the statistical results should be clearly marked in the chart or text to enhance the credibility of the data.
- Sensory evaluation is only based on 10 trained judges and three fermentation replicates, and the sample size is small, which may affect the universality of the results. It is suggested to increase the sample size or explain the representativeness of sample selection to improve the reliability of the conclusion.
- The explanation of microbial community and metabolite changes in the discussion is superficial, lacking of in-depth comparison with existing studies. For example, the difference of microbial composition between Tsounati and other olive varieties and its potential reasons have not been fully discussed. It is suggested to analyze the possible driving factors of these differences (such as geography, climate or processing conditions) with more literature.
Round 2
Reviewer 1 Report
Comments and Suggestions for Authors
According to the revised manuscript submitted by the author, the author has corrected the inaccurate descriptions, poor coherence, and other defects in the manuscript, and corrected most of the obvious problems. In the response to the reviewer's comments, the author answered the reviewer's questions in detail, had a meaningful exchange, and elaborated on the specific intentions of the controversial parts.
I think that the author's revisions to the manuscript have improved the quality of the manuscript and are close to perfection. It is recommended that the editorial team make a final decision after reviewing whether there are other problems with the manuscript.